



# Nitrogen deposition in the UK at 1 km resolution from 1990 to 2017

Samuel J. Tomlinson[1], Edward J. Carnell[2], Anthony J. Dore[2] and Ulrike Dragosits[2]

[1]UKCEH, Lancaster Environment Centre, Library Avenue, Bailrigg, LA1 4AP
[2]UKCEH, Bush Estate, Penicuik, Midlothian, EH26 0QB

*Correspondence to*: Samuel J. Tomlinson (samtom@ceh.ac.uk)

**Abstract.** An atmospheric chemistry transport model (FRAME) is used here to calculate the UK N deposition for the years 1990-2017. Reactive nitrogen (N) deposition is a threat that can lead to adverse effects on the environment and human health. In Europe, substantial reductions in N deposition from nitrogen oxide emissions have been achieved in recent decades, this
paper quantifies reductions in UK N deposition following the N emissions peak in 1990. In the UK, estimates of N deposition are typically available at a coarse spatial resolution (typically 5 km x 5 km grid resolution) and it is often difficult to compare estimates between years due to methodological changes in emission estimates. Through efforts to reduce emissions of N from industry, traffic, and agriculture, this study predicts that UK N deposition has reduced from 465 kt N in 1990 to 278 kt N in 2017. However, as part of this overall reduction, there are non-uniform changes for wet and dry deposition of reduced N ($NH_x$)
and oxidised N ($NO_y$). In 2017, it is estimated 59% of all N deposition is in the form of reduced N, a change from 35% in 1990. This dataset uses 28 years of emissions data from 1990 to 2017 to produce the first long-term dataset of 28 years of N deposition at 1 km x 1 km resolution in the UK.

## 1 Introduction

The emissions and subsequent atmospheric deposition of nitrogen (N) have a well-documented list of effects on the global and
local environment (e.g. Stevens et al., 2018). N deposition is associated with impacts on ecosystem biodiversity (Nowak et al., 2015; Payne et al., 2017), eutrophication (Greenwood et al., 2019), soil acidification (Aggenbach et al., 2017), changes in carbon stocks (Britton et al., 2019) and human health (Nowak et al., 2018).

These threats are driven by anthropogenic emissions of oxides of nitrogen ($NO_x$) from sources such as fuel combustion including from road transport, and emissions of ammonia ($NH_3$), to which agriculture contributes around 85% annually in the
UK (NAEI, 2019). Previous studies generally show total deposition of N in the UK peaked around 1990, following the peak in emissions. Fowler et al. (2004) estimate around 430 kt N was deposited to the UK in 1990, with a 54% proportion of reduced N (predominantly ammonia). Using newer data, the Review of Transboundary Air Pollution report (RoTAP, 2012) re-estimated the total N deposition budget for 1990 in the UK to be ca. 380 kt N and finally Levy et al. (2020) estimated 410 kt N deposited. Since the beginning of the 1990s, deposition has reduced as mitigation policies have sought to curb emissions of
nitrogenous compounds, predominantly $NH_3$ and $NO_x$, but has stabilised at around 300 kt N yr$^{-1}$ from ca. 2010.



In order to study the many effects of N deposition and its trends over time, there must be appropriately detailed and consistent deposition estimates to use, across time and space. N deposition data in the UK are typically available at a 5 km x 5 km resolution (e.g. Levy et al., 2020). It is very likely, however, that this relatively coarse spatial resolution smooths out significant

variation at higher resolutions, which could be useful for studying effects. Smart et al. (2020) highlight this point by exploring the variance of a 5 km x 5 km and 1 km x 1 km N deposition output from the same model run, within a 10km square. They found the variance within the 1 km x 1 km product to be up to four times higher than that of the 5 km x 5 km product (within the same 10km square).

Another facet of N deposition to consider is that of cumulative loading and whether the impacts develop over time, and whether

they develop linearly (Payne et al., 2019; Payne et al.,2020). Payne et al. (2019) showed that N deposition effects on sensitive habitats should not only take account of the most recent best estimate, but that cumulative N deposition should be considered, e.g. over a period of 30 years. To enable such an approach, it is necessary to have a suitable consistent N deposition data series available. In the past, time series were often constructed by piecing together historical products that were using the best knowledge and datasets available at the time, rather than a single time series where all model output years are produced with

consistent model input data from the latest back-cast inventory dataset, and with the same version of the model and calibration methodology.

This new dataset consists of 28 years of 1 km x 1 km resolution N deposition data on the UK terrestrial surface, from 1990 to 2017, using a consistent approach to inputs and model calibration. This has been made available as part of The ASSIST programme (Achieving Sustainable Agricultural Systems; see https://assist.ceh.ac.uk).

## 2. Data and Methods

### 2.1 Atmospheric Chemistry Transport Modelling

The Fine Resolution Multi-pollutant Exchange (FRAME) is an atmospheric chemistry transport model (ACTM) used to calculate annual deposition of reduced and oxidised nitrogen (N) over the United Kingdom. The model is fully described elsewhere (Aleksankina et al., 2018; Dore et al., 2012; Dore et al., 2016; Vieno et. Al., 2010; Singles et al., 1998) and only the

relevant information for this work is reported here. The domain of the model covers Europe at 50km x 50km to provide the boundary conditions for the UK model domain with a grid resolution of 1 km x 1 km. A column of air with depth 2500 m is used to represent the relevant atmospheric processes. The column of air is advected across the model domain from all edge grid points and all wind directions with an angular resolution of 1 degree.

Emission of gaseous pollutants, vertical diffusion, chemical transformation, wet, and dry removal processes take place within

the air column. The model has 33 vertical layers with thickness varying from 1 m at the surface to 100 m in the upper layers. The model requires input data of both diffuse and point source emissions of ammonia ($NH_3$), oxides of nitrogen ($NO_x$) and sulphur dioxide ($SO_2$) (Vieno et. al., 2010).

FRAME uses land-cover-specific deposition velocities to generate dry deposition for up to five land cover categories: woodland, low-growing semi-natural vegetation, improved grassland, arable and urban (Land Cover Map 2015; Rowland et al., 2017). The model uses different scavenging coefficients for soluble gases and particles and assumes constant drizzle for calculation of wet deposition. An annual precipitation map (Tanguy et al., 2019 and Walsh, 2012) is used to drive the spatial variation in wet removal rate.

The FRAME model used for this work uses long term radio sondes mean wind speed (Dore et. Al., 2006) for all the years included here (1990-2017). The wind frequency is derived from modelled data from the Weather and Research Forecast model (Skamarock et al., 2019). The wind frequency used here is keep constant to a 2001-2012 mean for the year 1990-2001, and the specific year afterwards (2001-2017).

The FRAME model, for both the European and British Isles domains, was run for each year from 1990 to 2017, using the corresponding emission and wind/rainfall data. The land cover was kept constant throughout. The FRAME model version used was 9.15.0.

## 2.2 Emissions Data

### 2.2.1 Data sources

Input data were extracted and processed from the most recently available national emission inventory submissions from both the UK and the Republic of Ireland (EMEP, 2019; E-PRTR, 2019; NAEI, 2019). Emissions for the European domain were taken from Convention on Long-Range Transboundary Air Pollution (CLRTAP) submissions (EMEP, 2019). For agricultural $NH_3$ emissions, the latest set of annual emission maps from 1990-2017 was used, as derived for the UK's national atmospheric emission inventory. This inventory work utilises annual activity data at the holding level from the devolved authorities in the UK, i.e. Defra (England), the Scottish Government (Scotland), Welsh Assembly (Wales) and Daera (Northern Ireland) (see Carnell et al. (2019) for details).

Emissions data are routinely made available via sectors (e.g. Energy Production) and to create a consistent structure for all data sources. $NO_x$ and $SO_2$ emissions were restructured into the eleven Selected Nomenclature for sources of Air Pollution (SNAP) sectors (Table 1), developed by the European Topic Centre on Air Emissions (ETC/AE). Given the dominance of agriculture in $NH_3$ emissions, the FRAME model requires agricultural data to be split into livestock fertiliser emissions, with all non-agricultural sources as one sector (see Sect. 2.1.3).

The SNAP system is used in the UK for the annual updates to the National Atmospheric Emissions Inventory (NAEI, 2019). This corresponds to the main area of interest for the deposition outputs, and the Irish and wider European emissions were reformatted to match that reporting system. Whilst the UK, Ireland and the collated European data all use the Nomenclature For Reporting system (NFR, ca. 240 sectors – EEA, 2019), the UK collate the fine resolution categories into SNAP sectors





whereas the latter two report via the aggregated Generalised/Gridded Nomenclature for Reporting (GNFR). Table 1 also shows

how these two aggregated reporting systems broadly relate to each other.

Table 1. Selected Nomenclature for sources of Air Pollution (SNAP) sectors for Emissions Inventory reporting as outlined by CORINAIR, alongside the Generalised/Gridded Nomenclature for Reporting (GNFR) sectors (broadly matched).

| SNAP sector | SNAP Definition | GNFR Sector |
|---|---|---|
| 1 | Combustion in Energy Production & Transformation | A_PublicPower |
| 2 | Combustion in Commercial, Institutional & Residential & Agriculture | C_OtherStationaryCombustion |
| 3 | Combustion in Industry | B_Industry |
| 4 | Production Processes | |
| 5 | Extraction & Distribution of Fossil Fuels | D_Fugitive |
| 6 | Solvent Use | E_Solvents |
| 7 | Road Transport | F_RoadTransport |
| 8 | Other Transport & Mobile Machinery | G_Shipping H_Aviation I_Offroad |
| 9 | Waste Treatment & Disposal | J_Waste |
| 10 | Agriculture Forestry & Land Use Change | K_AgriLivestock L_AgriOther |
| 11 | Nature | N_Natural |
| NA | Do not count towards national totals | O_AviationCruise P_IntlShipping |

It is worth noting that emissions data for International Shipping and Aviation Cruise do not count within a specific national inventory, but are reported into a 'pooled' total by all countries. Separate totals for national shipping, airports and the take-off and landing of aircraft are reported on a country basis. Finally, emissions data should ideally be translated between the aggregated classification systems using the NFR codes upon which they are built (which still has some one-to-many relationships) but spatial data are not available at this level and therefore the aggregated spatial data should not be broken

down in an attempt to make the NFR level data.



### 2.2.2 Point and diffuse emissions of NO$_x$, SO$_2$ and NH$_3$

NH$_3$, NO$_x$ and SO$_2$ emission inputs were produced for the years 1990 to 2017, for both diffuse and point source emissions. Diffuse sources are those deemed to be areal, non-exact locations such as agriculture, vehicles, population-related sources etc. Point sources can be located by exact coordinates, for example the actual chimney/exhaust stacks of power stations and industry (Vieno et al., 2010). Point source information in the UK is nearly (but not totally) exclusive to energy generation and industry.

Fig. 1 shows an overview of the processes to combine the various spatial and tabulated emissions data that are required for the 28 annual model runs. There are some important methodological details, for both diffuse and point emissions, worth noting. In the UK, diffuse data is produced and published for 11 SNAP sectors for the latest emissions inventory year, superseding any previous data. This is principally due to the fact that every year in the inventory compilation, minor to major changes are made to the way the data is compiled – this could be changes to emission factors with the latest research being incorporated or how underlying spatial methods and datasets are updated. While the non-spatial data are "back-cast" to 1990 (or earlier, depending on the pollutant), the maps are not currently updated as a time series. Consequently, it is unwise to compare previous years' gridded emissions surfaces to the latest available. For this reason, at the time of publication, only the latest 2017 emissions maps were used in the UK for the entire time series, and were scaled back through the time series using the tabulated NFR annual totals, for SO$_2$, NO$_x$ and non-agricultural NH$_3$. For agricultural NH$_3$, the latest mapped time series (using annual livestock and crop data) was used (Carnell et al. 2019). For point sources - which in the more recent data number in the thousands - some earlier data were obtained back to 1990 but only for a subset of major polluters and not for all years (missing years were linearly interpolated). For the very largest emitters, information (when known) regarding the stack/chimney height, stack/chimney diameter and emission exit velocities is also used by the model to create plume characteristics. It is the non-coordinate parameters that are important in determining to what height into the atmosphere the emissions travel, and therefore what subsequent chemical interactions occur, which is important for the deposition modelling.



Figure 1. Visualised methodology of steps to create inputs for the Fine Resolution Multi-pollutant Exchange (FRAME) atmospheric chemistry transport model; rectangle with corners missing (solid border) = spatial data, rectangle with corners missing (dashed border) = tabulated data, rectangle with rounded corners = process, oval = model.

Emissions from the Republic of Ireland influence the deposition of N species in the UK. To allow for similarly high resolution emissions inputs, the outputs from the National Mapping of GHG and non-GHG Emissions Sources project (MapEire, 2019;



Pjeldrup et al., 2018) were used in a similar manner to the latest emissions surfaces produced for the UK in the NAEI. The MapEire project produced 1 km x 1 km resolution gridded emissions for all GNFR sectors for the year 2016, which were scaled to other years by the totals reported to the CLRTAP by the Republic of Ireland. These surfaces were then transformed to SNAP sectors (see Table 2.) to be joined to the UK data. One important difference to note is that the MapEire gridded data

include all sources of emissions, including point sources (the UK data does not). Therefore, the major emitting point sources, as reported to the European Pollutant Release and Transfer Register (E-PRTR, 2019), were extracted for $NO_x$ and $SO_2$ for all available years back to 1990 (gaps were linearly interpolated). To conserve totals, Irish point values were removed from the Irish total gridded surface by subtracting the point value from the grid cell in which it was located, with any surplus emissions removed from the surrounding eight cells on an equal share basis (if required). This created a diffuse surface and a point source

input, consistent with the UK data.

A consistent time series of UK agricultural $NH_3$ emission estimates was created at a 1 km x 1 km grid resolution for the years 1990 – 2017. These high resolution agricultural $NH_3$ emission maps are produced annually for the NAEI, using an agricultural emission model jointly developed by the UK Centre for Ecology & Hydrology, Rothamsted Research, ADAS and Cranfield

University. The emission model uses annual agricultural census data (e.g. livestock numbers and crop areas – see Carnell et al., 2019) at the holding level, agricultural practice information (e.g. fertiliser application rates, stocking densities) and emission source strength data from the UK emissions inventories for agriculture (e.g. Brown et al. 2019; Richmond et al. 2019). Emission estimates are output for each individual emission source at a 10 km x 10 km grid resolution, which are spatially disaggregated to a 1 km x 1 km grid resolution using land cover data (Rowland et al., 2017) and methods outlined in Dragosits

et al. (1998), Hellsten et al. (2008) and Carnell et al. (2019). Emissions sources are numerous and include grazing, storage, spreading and housing for cattle, pigs, poultry, sheep and minor livestock (plus all sub-types), as well as differing fertiliser applications for varying crop and grass types.

## 2.3 Outputs

Outputs from the model as presented in this dataset are the annual values of wet and dry deposition of reduced nitrogen ('$NH_x$'), and wet and dry deposition of oxidised nitrogen ('$NO_y$') as a weighted mean of all land cover types within a given cell, as well as vegetation specific values to both forest and moorland – Table 2 provides more detail.

Table 2. Deposition outputs as provided in this dataset from the Fine Resolution Multi-pollutant Exchange (FRAME)

atmospheric chemistry transport model.

| Name | Long Name | Description | Units |
| --- | --- | --- | --- |



| NX$_x$ dry | Dry deposition of reduced N | Grid average deposition of NH$_3$ + NH$_4$, plus forest and moorland specific deposition | Kg N ha$^{-1}$ year$^{-1}$ |
|---|---|---|---|
| NH$_x$ wet | Wet deposition of reduced N | Grid average deposition of NH$_3$ + NH$_4$, plus forest and moorland specific deposition | Kg N ha$^{-1}$ year$^{-1}$ |
| NO$_y$ dry | Dry deposition of oxidised N | Grid average deposition of NO$_2$ + NO$_3$ + HNO$_3$ + PAN, plus forest and moorland specific deposition | Kg N ha$^{-1}$ year$^{-1}$ |
| NO$_y$ wet | Wet deposition of oxidised N | Grid average deposition of NO$_3$ + HNO$_3$, plus forest and moorland specific deposition | Kg N ha$^{-1}$ year$^{-1}$ |

Deposition data are provided on a 1 km x 1 km resolution surface, using the British National Grid projection (same domain as the emission files) for UK terrestrial cells (n. cells = 259,436). Other land cover types used in the calculations (but not output) are arable, urban and improved grassland.

## 2.4 Evaluation

### 2.4.1 Observation Data

ACTM results were evaluated using measured annual mean concentrations from rural background monitoring stations throughout the UK, via the UK Acidifying and Eutrophying Atmospheric Pollutants (UKEAP) network (UK AIR, 2020). Mean annual data were used if there was a data capture greater than 50% for a given site in a given year, which allows not only for direct comparison between modelled and measured data but also allows for a certain amount of smoothing of potential variability in the measured data due to natural factors (Chang & Hanna, 2004). Table 3 outlines the available measurement networks and the data they provide.

Table 3. Four measurement networks used within the UK Acidifying and Eutrophying Atmospheric Pollutants (UKEAP) network, along with the ten compounds used to evaluate the atmospheric modelling.

| Network | Long Name | Data Provided | Units |
|---|---|---|---|
| NAMN | National Ammonia Monitoring Network | NH$_3$ – Ammonia conc. in gas <br> NH$_4$ – Ammonium conc. in aerosol | µg m$^{-3}$ <br> µeq l$^{-1}$ |
| PrecipNet | Precipitation Network | NO$_3$ – Nitrate conc. in precipitation <br> NH$_4$ – Ammonium conc. in precipitation | µeq l$^{-1}$ <br> µeq l$^{-1}$ |
| Rural NO$_2$ | Rural Background NO$_2$ | NO$_2$ – Nitrogen Dioxide conc. in gas | µg m$^{-3}$ |



| AGANET | Acid Gases & Aerosol Network | NO$_3$ – Nitrate conc. in aersol | µg m$^{-3}$ |
| | | HNO$_3$ – Nitric acid conc. in gas | µg m$^{-3}$ |

**2.4.2 Evaluation Metrics**

It is unlikely for an ACTM to perfectly reproduce reality due to errors in, but not limited to, input data, model physics and chemistry schema, uncertainty in meteorological data and the random effects of the real world. However, using methods outlined in Chang & Hanna (2004), several statistical metrics may be used to evaluate the agreement between the modelled predictions and the real world observations; fraction of predictions within a factor of two of observations (FAC2), the fractional bias (FB), the normalized mean square error (NMSE) and the geometric mean bias (MG). These metrics are defined in the following way:

$$FAC2 = fraction\ of\ data\ that\ satisfy\ \ 0.5 \leq \frac{C_p}{C_o} \leq 2.0 \tag{1}$$

$$FB = \frac{(\overline{C_o} - \overline{C_p})}{0.5(\overline{C_o} + \overline{C_p})} \tag{2}$$

$$NMSE = \frac{\overline{(C_o - C_p)^2}}{\overline{C_o} \cdot \overline{C_p}} \tag{3}$$

$$MG = exp(\overline{lnC_o} - \overline{lnC_p}) \tag{4}$$

Where: Co are measured observations and Cp are model predictions, the former being paired with the latter spatially. A perfect reproduction of measurement data would have; FAC2 = 1, FB = 0, NMSE = 0 and MG = 1.

FAC2 is a robust measure of performance, not overly influenced by outliers, indicating the proportion of modelled/measured pairs falling within a factor of 2 of each other. FB is a linear metric that measures the mean systematic bias of the model and may have predictions out of phase with measurements but still return a value of 0 due to cancelling errors. NMSE is a measure of mean relative scatter and reflects both systematic and random errors. Finally MG, also a measure of mean systematic bias, but is less influenced by extreme values as it is a logarithmic metric (see Chang and Hanna (2004) for more detail). Hanna and Chang (2012) suggest that a model should satisfy at least 50% of the criterion used (two of four in this study), while the acceptability criterion for each metric are as defined in Theobald et al. (2016): FAC2 > 0.5, |FB| < 0.3, NMSE < 1.5 and 0.7 < MG < 1.3.



## 3. Results and Discussion

### 3.1 Emissions

In the UK, stricter air pollution policies, improving technology and changes in fuel use have all contributed to the reduction of emissions. Initially, mitigation strategies concentrated on $SO_2$ emissions, but the focus was extended to nitrogen compounds such as $NO_x$ (as well as VOCs) in an attempt to abate acidification and, latterly, to $NH_3$ (Grennfelt and Hov, 2005; Carnell et al., 2019). Within the model domain, emissions of $NH_3$ and $NO_x$ have decreased by ~12% and ~64% respectively from 1990 to 2017 (Fig. 2).



Figure 2. Emissions (in kt) of ammonia ($NH_3$), nitrogen oxides ($NO_x$) and sulphur dioxide ($SO_2$) in the model domain, covering
the UK and Ireland, from 1990 to 2017, split into the main broad reporting sectors.



Much of the decrease in emissions of $NO_x$ in the UK has been driven by the decline of coal use in power stations (95% decrease in emissions over the time series) and the improvement and modernisation of petrol combustion in road transport (98% decrease in emissions over the time series). Decreases in $NO_x$ have been offset by increases in emissions from DERV (diesel

fuels) and aviation fuels. With regard to $NH_3$ emissions, which are dominated by agriculture, changes in farm practices have seen a patchwork of decreases and increases to various emissions sources, with a generally decreasing trend that has plateaued from ca. 2001. It is the non-agricultural sources, however, that have shown marked increases from 1990 to 2017, including those activities associated with the circular economy; anaerobic digestion, composting of organic materials, application of sewage sludge to land and the combustion of biomass for industry (total increase; ~5kt to ~26kt). Finally, $SO_2$ emissions have

reduced by ~94% in the same time period (mean of ~5% yr-1), which is a direct result of the decline of coal use, especially in power stations, and restrictions being placed on the sulphur content of various fuels.

As all three pollutants are reactive in the atmosphere, differing rates of emissions reductions have varying effects on chemical reactions and subsequent deposition. Changes to emissions over time vary in space and so does, therefore, N deposition (Fowler et al., 2007).

**3.2 Model Evaluation**

Scatter plots of the modelled predictions vs measurements in 2017, for data collected in Table 3., are shown in Fig. 3. The associated performance metrics are given in Table 4.

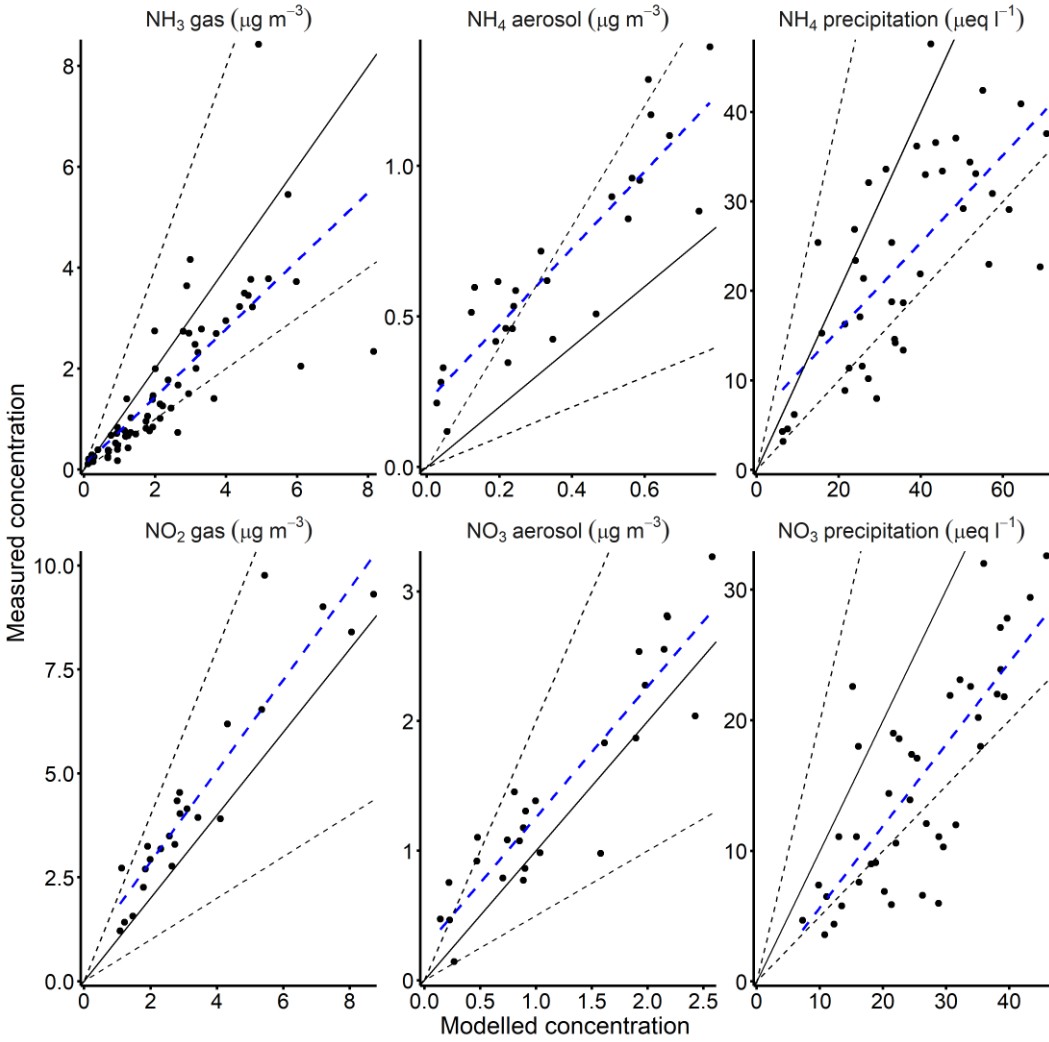

Figure 3: Evaluation of modelled (x-axis) and measured (y-axis) concentrations of six nitrogen compounds in the UK for 2017 (see Table 3 for definitions). The solid black line represents a 1:1 relationship, and the dotted lines represent a factor of two (FAC2) relationship, the blue dashed lines are linear regressions.

Table 4: Evaluation metrics of modelled concentrations of six nitrogen compounds in gas, aerosol and precipitation in the UK for 2017 (see Table 3 for definitions). Bold numbers represent where that metric has been satisfied (see Sect. 2.4.2 for metric definitions).

| Metric | Acceptability | $NH_3$ (conc. in gas) | $NH_4$ (conc. in aerosol) | $NH_4$ (conc. in precip.) | $NO_2$ (conc. in gas) | $NO_3$ (conc. in aerosol) | $NO_3$ (conc. in precip.) |
|---|---|---|---|---|---|---|---|
| Points (n) | NA | 68 | 26 | 41 | 24 | 26 | 41 |



| | | | | | | | |
|---|---|---|---|---|---|---|---|
| $R^2$ | | 0.61 | 0.79 | 0.51 | 0.87 | 0.84 | 0.61 |
| FAC2 | > 0.5 | **0.76** | **0.50** | **0.76** | **0.96** | **0.85** | **0.63** |
| \|FB\| | < 0.3 | 0.33 | 0.62 | 0.42 | **0.26** | **0.20** | 0.50 |
| NMSE | < 1.5 | **0.44** | **0.54** | **0.35** | **0.12** | **0.10** | **0.37** |
| MG | > 0.7 & < 1.3 | **0.70** | 2.29 | 0.64 | 1.33 | 1.31 | 0.56 |

For the latest year included in this study, all six N forms in Table 4 comply with the FAC2 metric and all six comply with the
recommended NMSE limit of 1.5. FB and MG are met with less success, though all are close to the recommended thresholds,
aside from $NH_4$ in aerosol (which contributes to dry deposition). FB and MG measure the systematic bias of the model and for
both $NH_4$ and $NO_3$, the model is slightly under-predicting the aerosol phase and over-predicting the aqueous phase. Not shown
in Fig. 3 and Table 4 is the evaluation of HNO3 in gas, which similarly fulfils recommendations for FAC2 (0.54) and NMSE
(0.48), but not for \|FB\| (0.48) or MG (0.56). N.B. Modelled predictions were also evaluated for 2016, with all seven compounds
achieving 50% compliance with $NH_3$ in gas, $NO_2$ in gas and $HNO_3$ in gas satisfying all four. It is not fully known why 2016
achieves better evaluation scores, it may be random variations in real world conditions, but one reason may be that 2017 was
a relatively warm year by annual mean temperature standards (and 4th warmest on record for England only). It is known that
$NH_3$ emissions are effected by temperature (e.g. Hempel et al., 2016, Sutton et al. 2013, Riddick et al. 2018) and, as temperature
fluctuations are not factored into the model or into the underlying emission inventories, this may have driven higher
spring/summer emissions of $NH_3$ and therefore higher dry deposition episodes.

This evaluation would indicate that total wet deposition was over-predicted and total dry deposition was under-predicted. To
provide further context and evaluation, measurement data were obtained for three previous years spanning the time series at
equal intervals; 1990, 1999 and 2008. Data for historic years, especially prior to ~1998, are limited and so scatter plots in Fig.
4 show the relationship between modelled predictions and measured data for four N compounds while Table 5. shows the
associated performance metrics.



Figure 4: Evaluation of modelled (x-axis) and measured (y-axis) concentrations of four nitrogen compounds in the UK for
1990, 1999 and 2008 (see Table 3 for definitions; no $NH_3$ gas data exist for 1990). The solid black line represents a 1:1
relationship, and the dotted lines represent a factor of two (FAC2) relationship, the blue, green and red dashed lines are linear
regressions.

Table 5: Evaluation metrics of modelled concentrations of six nitrogen compounds in gas, aerosol and precipitation in the UK
for (a) 1990, (b) 1999 and (c) 2008 (see Table 3 for definitions). Dashed lines represent no available data. Bold numbers
represent where that metric has been satisfied (see Sect. 2.4.2 for metric definitions).



| (a)  1990 | | NH$_3$ | NH$_4$ | NH$_4$ | NO$_2$ | NO$_3$ | NO$_3$ |
| Metric | Acceptability | (conc. in gas) | (conc. in aerosol) | (conc. in precip.) | (conc. in gas) | (conc. in aerosol) | (conc. in precip.) |
| --- | --- | --- | --- | --- | --- | --- | --- |
| R$^2$ | | - | - | 0.51 | 0.85 | 0.60 | - |
| FAC2 | > 0.5 | - | - | **0.69** | **1.00** | **0.40** | - |
| \|FB\| | < 0.3 | - | - | 0.44 | **0.14** | 0.73 | - |
| NMSE | < 1.5 | - | - | **0.45** | **0.11** | **0.81** | - |
| MG | > 0.7 & < 1.3 | - | - | 0.61 | **0.80** | 0.44 | - |

| (b)  1999 | | NH$_3$ | NH$_4$ | NH$_4$ | NO$_2$ | NO$_3$ | NO$_3$ |
| Metric | Acceptability | (conc. in gas) | (conc. in aerosol) | (conc. in precip.) | (conc. in gas) | (conc. in aerosol) | (conc. in precip.) |
| --- | --- | --- | --- | --- | --- | --- | --- |
| R$^2$ | | 0.29 | 0.66 | 0.63 | 0.77 | - | 0.66 |
| FAC2 | > 0.5 | **0.78** | **0.92** | **0.77** | **0.94** | - | **0.72** |
| \|FB\| | < 0.3 | **0.11** | **0.23** | 0.42 | **0.23** | - | 0.52 |
| NMSE | < 1.5 | **0.65** | **0.20** | **0.35** | **0.25** | - | **0.40** |
| MG | > 0.7 & < 1.3 | **1.03** | **0.88** | 0.66 | **0.78** | - | 0.58 |

| (c)  2008 | | NH$_3$ | NH$_4$ | NH$_4$ | NO$_2$ | NO$_3$ | NO$_3$ |
| Metric | Acceptability | (conc. in gas) | (conc. in aerosol) | (conc. in precip.) | (conc. in gas) | (conc. in aerosol) | (conc. in precip.) |
| --- | --- | --- | --- | --- | --- | --- | --- |
| R$^2$ | | 0.44 | 0.88 | 0.55 | 0.91 | 0.91 | 0.61 |
| FAC2 | > 0.5 | **0.82** | **0.88** | **0.81** | **1.00** | **0.93** | **0.57** |
| \|FB\| | < 0.3 | **0.02** | **0.07** | 0.34 | **0.09** | **0.29** | 0.56 |
| NMSE | < 1.5 | **0.54** | **0.08** | **0.33** | **0.10** | **0.20** | **0.47** |
| MG | > 0.7 & < 1.3 | **0.94** | **1.11** | **0.74** | **0.98** | **0.82** | 0.53 |

All N forms for which data were available in 1990, 1999 and 2008, satisfy at least two of the four evaluation metrics, with four gas and aerosol N compounds fulfilling all metrics in 2008. An example of the benefit of multiple evaluation metrics is shown in Fig. 4 when looking at NO$_2$ and NH$_3$ in gas in 2008. Both have very low FB values (indicating very low mean bias) due to the cancelling effect around the 1:1 line but the scatter of predictions to measurements of NH$_3$ is clearly much larger than for NO$_2$. Information of the NMSE and the FAC2, plus visual inspection of the plots, help to illustrate that NH$_3$ has a larger error 285 than NO$_2$.





### 3.3 Nitrogen Deposition

Grid average N deposition – $NH_x$ wet and dry, $NO_y$ wet and dry – is plotted in Fig. 5 at a 1 km x 1 km resolution over the UK terrestrial surface, for 2017. The total N deposition over the UK is 278.3 kt N ($\bar{x}$ = 10.7 kg N ha$^{-1}$ yr$^{-1}$, s.d. = 4.5 kg N ha$^{-1}$ yr$^{-1}$

), with a maximum of 74.3 kg N ha$^{-1}$ yr$^{-1}$. Such high deposition values are reasonably rare (n. cells > 30 kg N ha$^{-1}$ yr$^{-1}$ = 118; n. cells > 50 kg N ha$^{-1}$ yr$^{-1}$ = 8) and are a direct result of the increased resolution of the model, when compared to the maximum deposition of 5 km x 5 km resolution N deposition.

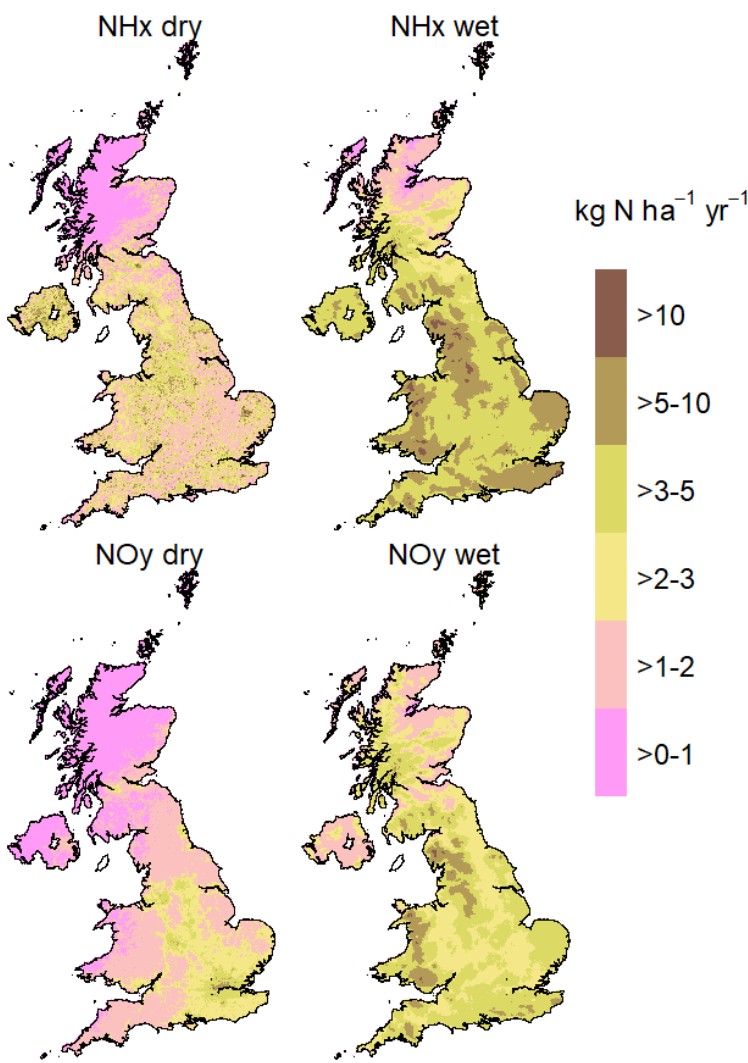

Figure 5. Four forms of nitrogen (N) deposition over the UK terrestrial surface in 2017 at 1 km x 1 km resolution, for grid
average land cover: wet/dry deposition of reduced N ($NH_x$) and wet/dry deposition of oxidised N ($NO_y$) (kg N ha$^{-1}$ yr$^{-1}$).



The two wet deposition surfaces in Fig. 5 exhibit smoother patterns (compared to dry deposition), a reflection of the precipitation surface across the UK, and constitute ~67% of the total deposition. It should be noted that, as shown in Figs. 3 and 4, deposition in precipitation of both $NH_4$ and $NO_3$ are consistently over predicted by the model throughout the time series.

Upland areas are subject to the highest values of wet deposition and most of the highest value cells between 25 and 50 kg total N ha$^{-1}$ yr$^{-1}$ are dominated by wet deposition. Dry deposition of $NO_y$, as modelled in this study, is the smallest contributor to total N deposition (~14%) and is dominated by $NO_2$ and $HNO_3$, which both follow their respective concentration fields closely (RoTAP, 2012). Dry deposition of $NO_2$, therefore, is largest in urban areas and close to road networks such as motorways. Dry deposition of $NH_x$, ~20% of total N deposition, is a highly heterogeneous surface with the highest values associated with areas

of intensive livestock farming (including beef, dairy, pigs and poultry). Gaseous $NH_3$ has a short atmospheric lifetime and so is deposited close to the sources. The very highest values of total N deposition (> 50 kg N ha$^{-1}$ yr$^{-1}$) are all dominated by dry deposition of $NH_x$ and are located near high agricultural emissions. An important factor in the deposition of $NH_x$ is the presence of oxidised $SO_2$, sulphuric acid ($H_2SO_4$), to form the aerosol $(NH_4)_2SO_4$. With decreasing $SO_2$ available to create $H_2SO_4$, more $NH_3$ is deposited within short distances as dry deposition. This effect is further enhanced by the increased rate of dry deposition

of the available $SO_2$, a result of the increase in the concentration ratio of $NH_3:SO_2$ which increases surface water pH, which further limits the available $SO_2$ to oxidise to $H_2SO_4$ (Baek & Aneja, 2004; Fowler et al., 2007; RoTAP, 2012; Tan et al., 2020).

Looking at the pattern of modelled N deposition from 1990 to 2017, Fig. 6 shows a steady decrease of wet and dry $NO_y$ deposition, a slow decrease of wet $NH_x$ deposition and no apparent decrease of dry $NH_x$ deposition. The latter is due to the

change in atmospheric chemistry with declining sulfur emissions due to successful policy implementation. Total N deposition over the UK has decreased from 465 kt N to 278 kt N, though no significant reductions in the total have occurred since around 2011.

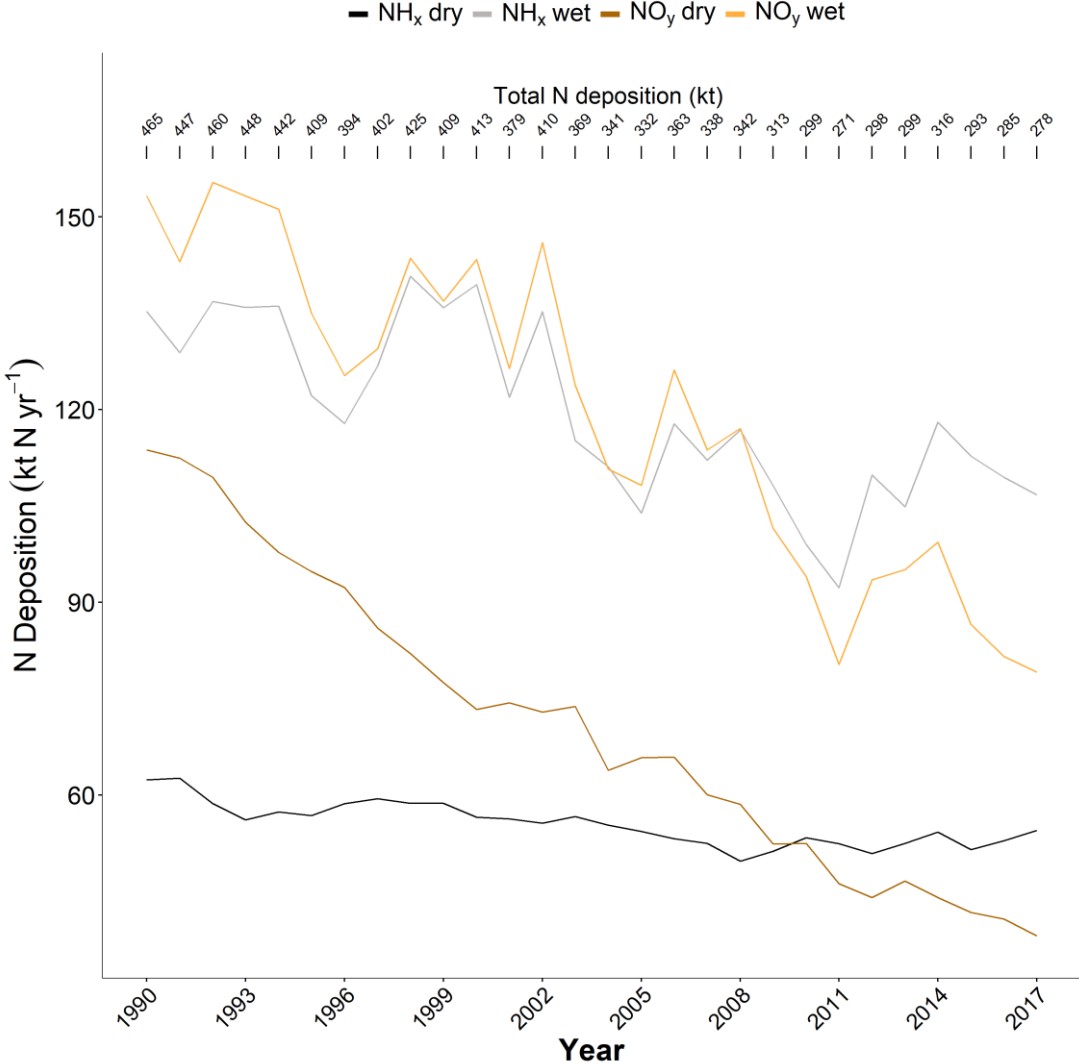

Figure 6. Four forms of total nitrogen (N) deposition over the UK terrestrial surface from 1990 to 2017, for grid average land
cover: total wet/dry deposition of reduced N (NH$_x$) and wet/dry deposition of oxidised N (NO$_y$) (kt N yr$^{-1}$).

Total oxidised N deposition has decreased by ~56% from 1990 to 2017, while reduced N deposition has decreased by ~19%.
This reflects the larger emissions reductions achieved for NO$_x$ than for NH$_3$ from 1990. Mean deposition values for all four N
forms have changed in a similar fashion to their respective totals from 1990, but the standard deviation across all 5 km x 5 km
cells for oxidised N (both wet and dry) has decreased over time, possibly due to the heavy reductions in emissions sources
such as road traffic and power stations, which previously created very high localised dry deposition. Figure 7 shows every year
of total N deposition from 1990 to 2017, and highlights the non-linear relationship between decreasing emissions and
deposition.












Figure 7. Spatial distribution of total nitrogen (N) deposition over the UK terrestrial surface, 1 km x 1 km resolution, from 1990 to 2017, for grid average land cover (kt N yr$^{-1}$).

Some of the areas with highest N deposition in later years are remote upland areas, which are principally effected by longer-

range wet deposition (and transboundary deposition) and have seen much lower relative decreases in N deposition than lowland areas such as southeast England. NO$_x$ emissions have decreased by ~64% across the time series, and resulting wet and dry NO$_y$ deposition decreases of ~48% and ~66%, respectively. This illustrates the non-linear processes involved with the chemical processing of NO$_x$ emissions, in particular the resulting concentrations of NO$_3$ in precipitation which are not decreasing at the same rate as gas and/or aerosol forms of oxidised N (see Fowler et al., 2007; Sickles and Shadwick, 2015; Feng et al., 2020).

It must be recognised again, however, that the model is over-estimating wet deposition of N to a degree.

As a result of emissions changes and non-linear chemistry, estimates of modelled dry deposition have decreased as a percentage of the total N deposition (1990 = ~38%, 2017 = ~33%) (see Fig 8.). This dataset models wet deposition as the dominant source of total N deposition.

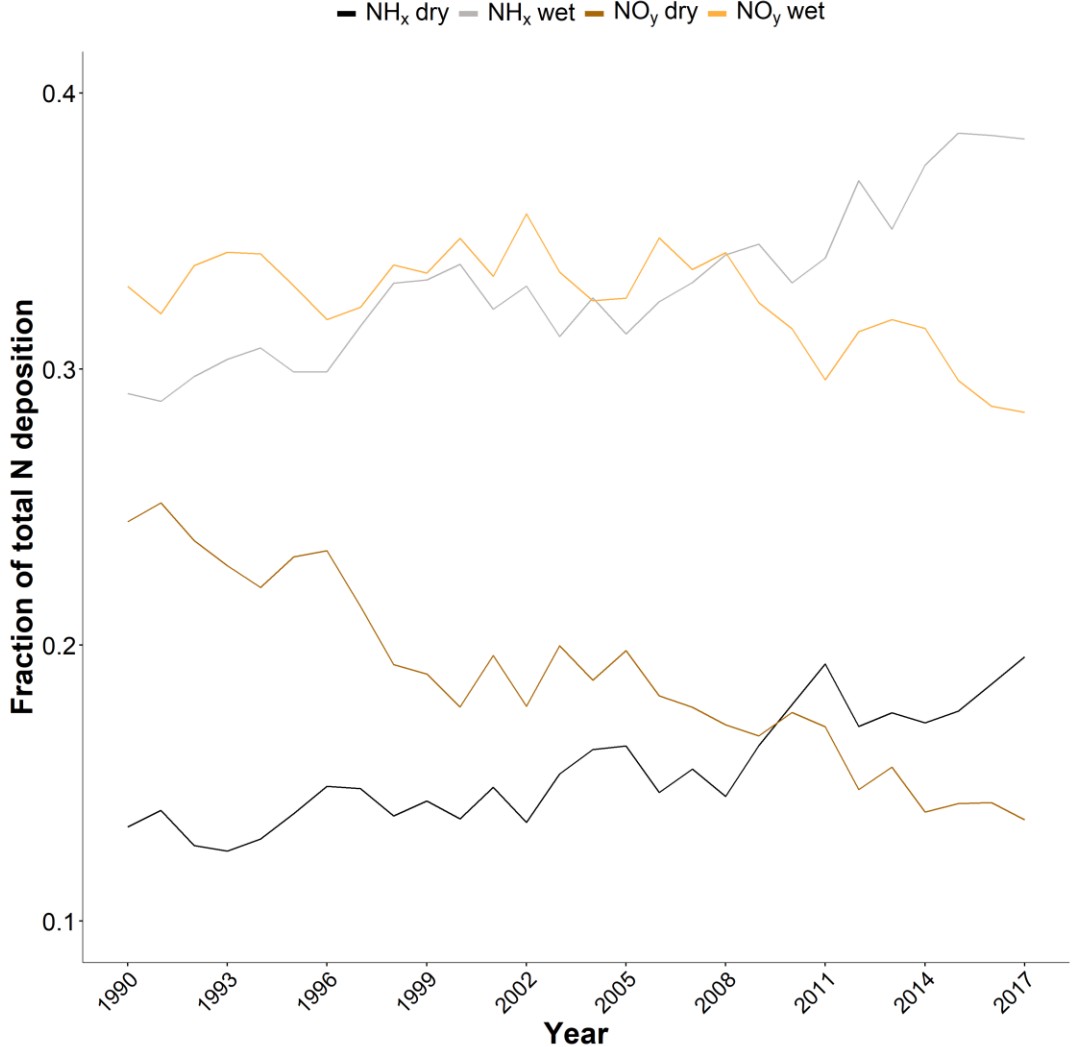

Figure 8. Fraction of the total nitrogen (N) deposition over the UK terrestrial surface for four forms of nitrogen (N) deposition, for grid average land cover, from 1990 to 2017: total wet/dry deposition of reduced N ($NH_x$) and wet/dry deposition of oxidised N ($NO_y$).

As a result of the large decreases of $NO_x$ emissions, and fewer regulations on most $NH_3$ emission sources in the UK compared to $NO_x$, reduced N is now the major component of N deposition. In this dataset, the proportion of dry deposition has moved from being dominated by oxidised N in 1990 (~65%) to reduced N in 2017 (~59%). This has resulted in a highly heterogeneous spatial distribution of N deposition that is more reflective of both agricultural practice and rainfall patterns.



## 4. Data Availability

The deposition data described in this paper are made available via the NERC Environmental Information Data Centre at https://doi.org/10.5285/9b203324-6b37-4e91-b028-e073b197fb9f (Tomlinson et al., 2020).

## 5. Conclusions

This new dataset provides a consistent time series of modelled wet and dry deposition of both reduced and oxidised N (plus
total N) for the whole UK terrestrial surface on a 1 km x 1 km resolution (n. cells = 259,436), from 1990 to 2017. Atmospheric modelling was undertaken for all 28 years and there is good agreement between modelled predictions and measured observations of various compounds of N not only for 2016 and 2017, but also selected prior years where tests were carried out (1990, 1999 and 2008). It is estimated within this dataset that N deposition has undergone large decreases across the time period, from 465 kt N to 278 kt N, but that a cessation in the decrease of $NH_3$ emissions (plus vast reductions in $SO_2$ emissions)
has seen reduced N become the dominant fraction of all N deposition. Higher resolution data enable more detailed effects studies across a wide range of disciplines, as well as cumulative effects from the annual time series. Further work should be aimed at improving the long-term spatial distribution of emissions.

## 6. Author Contributions

SJT designed and coded the methodology to combine all data sources into compatible input data, QAQC work, compiled a time series of rainfall data, reformatted FRAME Europe outputs to FRAME Europe inputs and analysed all model outputs, including historic model performance. EJC performed all of the agricultural emissions mapping for the time series, updated the FRAME UK land use files and undertook QAQC work. AJD undertook all atmospheric modelling requirements, principally the model runs and most recent evaluations. MV provided expert knowledge and advice with regard to atmospheric chemistry
and modelling. UD managed the atmospheric modelling task and offered expert advice on the spatial distribution of emissions and N deposition. SJT prepared the manuscript with contributions from all co-authors. All co-authors commented on the draft manuscript.

## 7. Competing Interests

The authors declare that they have no conflict of interest





**8. Acknowledgements**

This research was funded by the Natural Environment Research Council (NERC) under research programme NE/N018125/1
ASSIST – Achieving Sustainable Agricultural Systems www.assist.ceh.ac.uk. ASSIST is an initiative jointly supported by
NERC and the Biotechnology and Biological Sciences Research Council (BBSRC).

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

Met Éireann, Dublin.