# Peer review of "Nitrogen deposition in the UK at 1 km resolution from 1990 to 2017"

_Earth System Science Data, 2021_

## Author Response (AR1)

**https://essd.copernicus.org/preprints/essd-2021-112/essd-2021-112.pdf**

**Nitrogen deposition in the UK at 1 km resolution from 1990 to 2017**

**Reviewer 1:**

*This paper presents the nitrogen deposition in the UK at 1 km resolution from 1990 to 2017, and I believe the data will be useful for potential readers in the future. After I read the manuscript, I have some major concerns.*

We thank the reviewer for their comments, which have helped improved the quality of the manuscript. We responded to each comment below.

**There are several other estimates of N deposition in the UK, what are the new findings of your data? Just higher resolution? Or you used new model? Or you used more original data? This needs to be clarified, i.e., you used new data and new model, and then you created this new dataset, and your data are reliable.**

We believe this dataset represents the first time N deposition data in the UK has been made at a 1km x 1km resolution and over such a long time period. The model is not new (as described in Section 2.1) but its application in this way is. The following text was added to L48 to clarify this point,

"It is the first time annual N deposition data has been released at this resolution over this number of years in the UK, using a consistent methodology throughout. The consistent methodology means that the latest knowledge for emission distributions across the whole period can be used, with the latest emission factors used to back cast the entire time series at a high spatial resolution. In addition, model parameters and calibrations for each time step use the same most up-to-date model version. It is envisaged that studying the effects of N deposition on the environment can be aided by such an increase in detail, as suggested by Hallsworth et al. (2010)"

**L35-39, please point out the possible reasons for this, so readers can know how you fixed this.**

The authors have added the following text to the paragraph that ends on L38;

"The main driver for this increased variance of N deposition at higher spatial resolutions, compared to lower resolutions (within the same study area), is the more granular representation of dry N deposition from agricultural sources such as livestock houses, and busy roads or local combustion sources. Dry deposition of N from reduced nitrogen ('$NH_x$') is very local to the emissions sources which a 1km x 1km resolution can more easily reflect. Furthermore, but to a lesser extent, the increased definition in a 1km x 1km rainfall map (for wet deposition) has more variation than a smoothed 5km x 5km rainfall map, while land cover is more readily represented in higher resolutions (which can determine deposition velocities and therefore N deposition)."

**For the data evaluation, the authors presented the linear regressions for the six nitrogen compounds in the UK (Figure 3 and Figure 4). What are the accuracies of previous modelling data (for example, 10*10km resolution)?**

The authors agree that some more context regarding the performance of the correlations is important, and the following text has been added after L260;

"For context, Carslaw (2011) undertook a model inter-comparison exercise for the UK Department for Environment, Food & Rural Affairs (Defra), with a specific focus on deposition from the CMAQ, EMEP4UK, FRAME, HARM and NAME models. Respectively, those models (at the time) were run at resolutions of 12 km, 5 km, 5 km, 10 km and 12 km. The models reviewed by Carslaw (2011) performed with a similar correlation coefficient ('r') for all N compounds, aside from $NH_4$ and $NO_3$ in precipitation, for which the 2017 model run in this study had a weaker correlation (0.51 – 0.61 compared to 0.7 – 0.88)."

**I think it is necessary to present a map about the distribution of the observation sites.**

The authors agree and a new figure, Fig 3., has been inserted into Section 2.4.1, showing the locations of the observation sites for the four networks, across four specific years (that were evaluated in Section 3.2). Additional accompanying text was included immediately above Table 3:

"[…], while Fig 3. shows the spatial distribution of the observation sites with measurements in 1990, 1999, 2008 and 2017 (the first year of measurements for each observation network is noted in Table 3). It is believed that this is the first time model evaluation for gases, aerosols and concentration in precipitation has been done across a long time series at multiple points in time on the same dataset."

**Figure 8. It is difficult to distinguish the lines with different colors, please add the fractions near the lines.**

We agree, and Figure 8, now Figure 10 in the reviewed manuscript, has been changed to show the N fraction label close to the line that represents it.

**Reviewer 2:**

*N deposition is very crucial and is associated with ecosystem biodiversity and human activities. The dataset showed that the N deposition decreased for 1990-2017 in the UK. The method used in this article is reasonable and easy to understand. Having professed my general enthusiasm for the dataset and its importance, I have some concerns that some issues need to be addressed before it can be considered for publication.*

**There are several similar other N deposition data. What are new findings on this study? Can you do some contrastive analyses to show the improvement of your new data?**

We believe that the main strengths of this new dataset lie in the increased spatial resolution (I.e. 1km x 1km instead of 5km or 10km), the increased temporal scale (I.e. annual back to 1990), the consistency of modelling for the time series in terms of emissions processing (to avoid 'chaining' differing data together, or interpolating gaps) and the successful evaluation against measurements of four selected years across the entire time period, not just a focus on the more recent years.

The following text was added to L48 to clarify this point, also covering comments made by Reviewer #1 on the same topic:

"It is the first time annual N deposition data has been released at this resolution over this number of years in the UK, using a consistent methodology throughout. The consistent methodology means that the latest knowledge for emission distributions across the whole period can be used, with the latest emission factors used to back cast the entire time series at a high spatial resolution. In addition, model parameters and calibrations for each time step use the same most up-to-date model version. It is envisaged that studying the effects of N deposition on the environment can be aided by such an increase in detail, as suggested by Hallsworth et al. (2010)"

Re contrastive analyses, the authors agree some context regarding the performance of the correlations is important, and the following text has been added after L260;
We agree, and our changes also respond to similar queries from Reviewer #1.
For context, Carslaw (2011) undertook a model inter-comparison exercise for the UK Department for Environment, Food & Rural Affairs (Defra), with a specific focus on deposition from the CMAQ, EMEP4UK, FRAME, HARM and NAME models. Respectively, those models (at the time) were run at resolutions of 12 km, 5 km, 5 km, 10 km and 12 km. The models reviewed by Carslaw (2011) performed with a similar correlation coefficient ('r') for all N compounds, aside from $NH_4$ and $NO_3$ in precipitation, for which the 2017 model run in this study had a weaker correlation (0.51 – 0.61 compared to 0.7 – 0.88).

**The paper did evaluation with four different metrics. None of these N depositions pass all four metrics (Table 4). But the results look better while separating the data into three parts (Table 5). How will these evaluation metrics have affected by degree of freedom? Are these evaluation metrics good enough or still need to improve?**

While none of the forms of N pass all four nominated metrics in 2017, the requirement of 50% of metrics to be met is adhered to. As mentioned in L255, the model performs slightly better when evaluated against 2016 measurements, meaning there are some inter-annual fluctuations in non-modelled conditions (such as climatic differences) that may account for some of the evaluation metrics not being met. These are the most common and widely accepted metrics used for atmospheric modelling and while this list could be more exhaustive, the authors feel the four chosen, along with the r value as an indicative 5[th], is sufficient evidence.
There may be some misunderstanding by the reviewer, but Table 4 displays evaluations for the latest modelled year, 2017, while Table 5 displays the exact same evaluation but for 3 other selected years (1990, 1999 and 2008) in an attempt to draw out some statistical conclusions for the model performance across the time series, instead of just the latest year. In table 5, the amount of data points for the N form has been added.

**Did you do the sensitivity analysis to address the uncertainties of your model to make sure the results reliable?**

A sensitivity analysis was not undertaken in this study, however a previous sensitivity analysis on this model can be obtained from Aleksankina et al. (2018). The following text was appended to L285, to clarify;

"From the perspective of model sensitivity and/or uncertainty, there were no further model runs made with adjusted emissions inputs or adjusted deposition parameters within this study. However, Aleksankina et al. (2018) employed statistical techniques to obtain uncertainty estimates of the FRAME model, representing model runs with a ±40 % variation range for the UK emissions of $SO_2$, $NO_x$, and $NH_3$. They found that the sensitivity of concentrations of primary

precursors $NO_x$ and $NH_3$, plus the deposition of N, were dominated by emissions. However, concentrations of secondary species such as particulate $NO_3^-$ and $NH_4^+$ were more geographically dependent."

**What are the temporal-spatial resolution of measured observations used in the evaluations?**

We have added two columns into Table 3, 'Measurement Resolution' and 'Start Year' to indicate the temporal scale and resolution of measurements for each network. Furthermore a new figure, Fig 3., has been inserted into Section 2.4.1, showing the locations of the observation sites for the four networks, across four specific years (that are evaluated in Section 3.2). The following text was added to the end of L177, and also responds to a similar comment from Reviewer #1:

[…]", while Fig 3. shows the spatial distribution of the observation sites with measurements in 1990, 1999, 2008 and 2017 (the first year of measurements for each observation network is noted in Table 3). It is believed that this is the first time model evaluation for gases, aerosols and concentration in precipitation has been done across a long time series at multiple points in time on the same dataset."

**There are huge heterogeneities in left-top panels of Figure 5. The authors explained this in Line 304-305. Please explain the heterogeneity difference in Figure5, i.e., why the side-by-side noises only exist in left-top panel not the other three?**

As the reviewer notes, heterogeneity in the top left of figure 5 (now Figure 7, NHx dry deposition) is expanded upon in the text. We feel the same is done for NOy dry deposition in L301-L303. The increased smoothness in the two wet deposition panels had been mentioned in L297 in the original manuscript, but we have changed the text as follows;
"The two wet deposition surfaces in Fig. 7 exhibit smoother spatial distributions and less heterogeneity (compared to dry deposition), a reflection of the precipitation surface across the UK, and constitute ~67% of the total deposition. Wet deposition is nearly always of a longer range than dry deposition, due to the transport in more elevated atmospheric layers, but some enhanced local washout around strong sources is also represented. This longer range transport acts as a smoothing effect on the deposition field due to the increased distance from the emission source."

**It is better to calculate the decrease/increase rates in Figure6&8 in addition to just showing the changing percentages.**

We appreciate the reviewer's comment but wonder if they misunderstood. We believe that we are already showing both absolute change (Figure 6, now Fig 8)) and relative change (Figure 8, now Fig 10).
Fig 6 (now fig 8) shows total N in kt across the years across all 4 types, while Fig 8 (now Fig 10) is the % of total N dep that each of the four forms is.

**It is better to show the latitude and longitude line in the geographic maps (Figure 5&7).**

We agree that more context should be given to the modelling domain in which the results are produced.

A new figure, Figure 1, has been produced to show the data created in the European context with lines of latitude and longitude. However, the mapped figures are produced in British National Grid, a projected coordinate system specific to the United Kingdom, which differs to the World Geodetic geographic coordinate system (commonly known as 'Lat-Long') in that it projects latitude and longitude onto a two-dimensional surface. We feel that it is correct to show the results in the coordinate system in which they were made and in which they are going to be most useful for further analysis, and which are relevant to the study area.

The following text is appended to L58;

"Fig. 1 shows the 1km x 1km UK model domain - which captures both the UK and the Republic of Ireland to allow for high resolution modelling of the closest neighbouring territory - in the European context. Further figures in this work di not show lines of latitude or longitude."